# Targeted Delivery Prodigiosin to Choriocarcinoma by Peptide-Guided Dendrigraft Poly-l-lysines Nanoparticles

**DOI:** 10.3390/ijms20215458

**Published:** 2019-11-01

**Authors:** Kai Zhao, Dan Li, Guogang Cheng, Baozhen Zhang, Jinyu Han, Jie Chen, Baobei Wang, Mengxia Li, Tianxia Xiao, Jian Zhang, Dongpo Zhou, Zheng Jin, Xiujun Fan

**Affiliations:** 1Engineering Research Center of Agricultural Microbiology Technology, Ministry of Education, Heilongjiang University, Harbin 150080, China; docor1005@163.com (D.L.); zhoudp0451@163.com (D.Z.); 2Key Laboratory of Microbiology, College of Heilongjiang Province, School of Life Science, Heilongjiang University, Harbin 150080, China; gangzi350274360@126.com (G.C.); jy.han1@siat.ac.cn (J.H.); 3Guangdong Key Laboratory of Nanomedicine, Center for Reproduction and Health Development, Institute of Biomedicine and Biotechnology, Shenzhen Institutes of Advanced Technology, Chinese Academy of Sciences, Shenzhen 518052, China; bz.zhang2@siat.ac.cn (B.Z.); jie.chen@siat.ac.cn (J.C.); bb.wang@siat.ac.cn (B.W.); limx@siat.ac.cn (M.L.); tx.xiao@siat.ac.cn (T.X.); jian.zhang@siat.ac.cn (J.Z.); 4School of Pharmaceutical Sciences, Key Laboratory of Bioorganic Phosphorus Chemistry & Chemical Biology (Ministry of Education), Beijing Advanced Innovation Center for Structural Biology, Tsinghua University, Beijing 100084, China; 5Key Laboratory of Chemical Engineering Process and Technology for High-efficiency Conversion, College of Chemistry and Material Sciences, Heilongjiang University, Harbin 150080, China; jinzhengdvd@163.com

**Keywords:** dendrigraft poly-l-lysines nanoparticles, placenta chondroitin sulfate A binding peptide, *Serratia marcescens* prodigiosin, targeted delivery, choriocarcinoma

## Abstract

The available and effective therapeutic means to treat choriocarcinoma is seriously lacking, mainly due to the toxic effects caused by chemotherapy and radiotherapy. Accordingly, we developed a method for targeting delivery of chemotherapeutical drugs only to cancer cells, not normal cells, *in vivo*, by using a synthetic placental chondroitin sulfate (CSA)-binding peptide (plCSA-BP) derived from malarial protein VAR2CSA. A 28 amino acids placental CSA-binding peptide (plCSA-BP) from the VAR2CSA was synthesized as a guiding peptide for tumor-targeting delivery, dendrigraft poly-L-lysines (DGL) was modified with plCSA-BP and served as a novel targeted delivery carrier. Choriocarcinoma was selected to test the effect of targeted delivery carrier, and prodigiosin isolated from *Serratia marcescens* subsp. *lawsoniana* was selected as a chemotherapeutical drug and encapsulated in the DGL modified by the plCSA-BP nanoparticles (DGL/CSA-PNPs). DGL/CSA-PNPs had a sustained slow-release feature at pH 7.4, which could specifically bind to the JEG3 cells and exhibited better anticancer activity than that of the controls. The DGL/CSA-PNPs induced the apoptosis of JEG3 cells through caspase-3 and the P53 signaling pathway. DGL/CSA-PNPs can be used as an excellent targeted delivery carrier for anticancer drugs, and the prodigiosin could be an alternative chemotherapeutical drug for choriocarcinoma.

## 1. Introduction

Cancer has been a major threat to human health in recent years [1,2,3]. In the past forty years, even though great efforts have been made, there is still much to be done before a safe therapy is achieved [4,5]. Currently, chemotherapy and radiotherapy are still the main therapeutic methods for cancer. However, most of the anticancer drugs have toxic side effects. Therefore, it is critical to develop novel anticancer drugs that have no toxic side effects on other tissues or organs in vivo, and only target the lesion area. Compared with traditional chemotherapeutic drugs, the nano-anticancer drug therapy has potential application prospects in curing cancer diseases [6,7].

One of the major concerns about the nano-anticancer drugs is the drug-delivery vector. The ideal delivery vector should possess such characteristics as a high drug-loading capacity, low immunogenicity, low toxicity, better shelf-life, and water solubility, and it should have the potential to be further modified [8,9]. Synthetic polymers, such as, chitosan, polyetherimide (PEI), poly(lactic-co-glycolic acid) (PLGA), polyamidoamine (PAMAM), and poly-l-lactic acid (PLA) have been widely used in gene- and drug-delivery systems [10,11,12]. However, stability of sustained-release formulations in aqueous solution, long delivery times, and low delivery efficiency are only a small part of the challenges highlighted by researchers [13]. Dendrigraft poly-l-lysines (DGL) have emerged as a new kind of synthetic polymer consisting of lysine [14,15,16] and have been employed as drug- or gene-delivery carriers, due to their biodegradability and rich external amino groups that can encapsulate drugs by the emulsion crosslinking method or plasmid DNA through electric interactions [9]. Besides their biodegradability [16], they can also be modified with targeting ligands and polyethylene glycol (PEG), thereby rendering vectors with targeting properties and long circulation. As yet, there are only a few ways available for targeted delivery of anticancer drugs. Recently, a report showed that VAR2CSA, which is exposed on the membrane of *Plasmodium falciparum* infected red blood cells (iRBC), could bind to 90% of tumors, which was verified by running immunohistochemistry on a tumor-tissue array [17]. Therefore, VAR2CSA is becoming an attractive target for anticancer drugs’ development. *P. falciparum* infection during pregnancy results in the sequestration of iRBC in the placenta by adhering to a distinct type of chondroitin sulfate A (CSA) exclusively expressed on trophoblast via VAR2CSA [18]. Short continuous regions of VAR2CSA with affinity for multiple types of CSA were defined [19]. A 28 amino acids placenta CSA-binding peptide (plCSA-BP) from the VAR2CSA region was recently tested as a guide peptide for targeted delivery of doxorubicin to choriocarcinoma [20]. However, whether it could be used to modify the DGL and its bioactivity after the modification was not evaluated.

The immunosuppressive, antimalarial, and pro-cytotoxic bacterial prodigiosin was recently described as a potent antimetastatic and anticancer agent [21,22]. It can induce apoptosis in a wide range of cancer cell lines, including hematopoietic cancer, breast cancer, gastric cancer, colon cancer, lung cancer, and rat hepatocellular carcinoma cell lines, with no marked toxicity in nonmalignant cells. However, prodigiosin has not been reported for the treatment of choriocarcinoma yet. It is selective in promoting apoptosis of malignant tumor cells, rendering prodigiosin a promising anticancer agent [23]. The antitumor mechanism of prodigiosin is still unclear, especially in regard to newly acquired prodigiosin produced by a small number of microorganisms, including *Serratia marcescens* and other bacteria, such as *Streptomyces griseoviridis*, *Serratia* spp., *Zooshikellarubidus, Vibrio* sp., *Actinomycetes*, and *Hahella chejuensis* [24,25,26]. Research on the mechanism of bacterial origin prodigiosin is very important for clinical application and drug development.

In this study, to develop a new type of targeted drug delivery carrier material and investigate the anticancer mechanism of prodigiosin, we first used plCSA-BP as a guide peptide to coat on the DGL vector for cancer-cell-targeted delivery of prodigiosin produced from *Serratia marcescens* subsp. *lawsoniana* HDZK-BYSB107. Then, we tested the binding ability of the newly developed DGL/CSA-PNPs and its anticancer effect in vitro on a JEG3 choriocarcinoma cell line and *in vivo* on a tumor model, respectively. In addition, the mechanism responsible for the anticancer effect of targeted nano-prodigiosin was also investigated. Overall, these findings suggested that the DGL/CSA vector is an excellent cancer-specific-drug delivery carrier, which has great potential as a novel delivery system for the treatment of cancer.

## 2. Results

### 2.1. Characterization of the Targeted DGL/CSA-PNPs and DGL/SCR-PNPs

Compared to the DGL (Figure 1A) and DGL-PEG-CSANPs (Figure 1B), the DGL/CSA-PNPs (Figure 1C) showed a spherical and polydisperse nature, as revealed by TEM. The DGL/CSA-PNPs had a regular round shape, good dispersion, and smooth surfaces. The average particle size of diameter of DGL/CSA-PNPs was 396.10 ± 13.27 nm (Figure 1D), and the zeta potential of DGL/CSA-PNPs was −(6.94 ± 0.65) mV (Figure 1E). The LC and EE of DGL/CSA-PNPs was 41.36 ± 0.87% and 89.39 ± 1.83% respectively, and higher than those of the DGL-PNPs and DGL/SCR-PNPs (Table 1).

### 2.2. In Vitro Release Assay of DGL/CSA-PNPs

As shown in Figure 1F, the bacterial prodigiosin release from DGL/CSA-PNPs at pH = 5.3 was much faster than that at pH = 7.4. The bacterial prodigiosin was burst-released at pH = 5.3, and the release amount reached 78.63 ± 0.82 %, within 12 h, at pH = 5.3. While the amount of prodigiosin was increased to 53.48 ± 1.53% from 0 to 12 h, at pH = 7.4, which reveals that the burst-release mainly takes place during the first 12 h, followed by a sustained slow release until 72 h, and reached 74.83 ± 0.75% at 72 h. In vitro release results showed that the DGL/CSA-NPs could be used as a delivery carrier of drugs for sustained release at pH = 7.4.

### 2.3. In Vivo Safety and In Vitro Cytotoxicity of DGL Formulations

To test the safety of DGL as carrier for the delivery of drugs, *in vitro* and *in vivo* cytotoxicity assays were performed. H&E staining showed that the concentrations of DGL from 50 to 1000 μg/kg have no significantly histopathological changes in lung, kidney, and liver within 30 days (Figure 2).

The CCK-8 assay indicated that the cell viability of DGL (100 μg/mL) can reach 81.62 ± 0.83%, the cytotoxicity of three kinds of DGL formulations at less than 10 μg/mL was safe (Appendix A). Meanwhile, the cell viability at 10 μg/mL showed no significant difference among the treated groups.

### 2.4. In Vitro Targeting Analysis of the Nanoparticles

To investigate the internalization of plCSA-BP upon binding to tumor cells, BODIPY was used as a tracer molecule and formulated in DGL/CSA-NPs to formulate DGL/CSA-BNPs, and the nontargeted DGL/SCR-BNPs were used as the control. JEG3 cells were cultured at 4 °C to exclude autonomous endocytosis, and green fluorescence can be seen in DGL/CSA-BNPs treated JEG3 cells (Figure 3A). However, no green fluorescence enrichment of BODIPY can be seen both in the free BODIPY and the DGL/SCR-BNPs-treated JEG3 cells. Thus, our results suggested that the plCSA-BP-loaded nanoparticles of DGL/SCR-BNPs can specifically target JEG3 cells.

### 2.5. In Vitro Cellular Uptake Assay of the Nanoparticles

As shown in Figure 3B, the uptake of BODIPY was significantly higher in JEG3 cells treated with the DGL/CSA-BNPs than in JEG3 cells treated with the DGL/SCR-BNPs or free BODIPY (*P* < 0.001), and the cellular uptake amount of the DGL/CSA-BNPs group was close to twice that of the DGL/SCR-BNPs or free BODIPY groups (Figure 3C). The findings further demonstrate that the DGL/CSA-NPs can effectively target the JEG3 cells.

### 2.6. In Vivo Antitumor Efficacy of the DGL/CSA-PNPs

To evaluate the antitumor efficacy of the targeted DGL/CSA-PNPs, a mice-tumor model based on the JEG3 cell lines expressed firefly luciferase was applied. As shown in Figure 4A,B, compared to the physiological saline, free prodigiosin, DGL-PNPs, and DGL/SCR-PNPs groups, the luciferase signal intensity of tumor in DGL/CSA-PNPs groups was significantly lower (*P* < 0.001). Simultaneously, the growth of the JEG3 tumor was significantly inhibited in the nude mice treated with the DGL/CSA-PNPs (*P* < 0.01) (Figure 4A,C,D), and the tumor size in the DGL/CSA-PNPs group did not exceed 200 mm^3^ until 18 days, while, in the other groups, it was more than 200 mm^3^ after the treatment for 16 days. Remarkably, over 70% mice in the DGL/CSA-PNPs group survived compared to less than 25% survival rate in other groups after the treatment for 18 days (Figure 4E). In summary, these results showed that the DGL-based targeted nano-delivery systems for drugs could significantly inhibit the growth of cancer cells and reduce the tumor volume. Additionally, the distribution and localization of CSA-guided nanoparticles in tumor-bearing mice were investigated in our previous work [20].

### 2.7. Apoptosis Induced by the HDZK-BYSB107 Prodigiosin and Prodigiosin Standard

To validate that the HDZK-BYSB107 prodigiosin has similar effects as the prodigiosin standard, the treated JEG3 cells were analyzed by using an Annexin V-propidium iodide (PI) apoptosis detection kit, according to the manufacturer’s instructions. The patterns of dose-response changes of annexin V/PI staining profiles of HDZK-BYSB107-prodigiosin-treated JEG3 cells were similar to those of the prodigiosin standard (Appendix A). By calculating the percentages of annexin V-positive (%), the apoptosis level shows a significant increase in a dose-dependent manner in both HDZK-BYSB107 prodigiosin-treated or prodigiosin-standard-treated JEG3 cells. The superior apoptosis effect for the HDZK-BYSB107 prodigiosin group was consistent with the prodigiosin-standard-treatment group. These results showed that we had obtained the high purity HDZK-BYSB107 prodigiosin, which has a similar ability of inducing apoptosis in cancer cells as the prodigiosin standard.

To further investigate the induction of apoptosis or other forms of cell death in JEG3 tumor obtained from the nude mice injected with the DGL/CSA-PNPs for 18 days, some important characteristic parameters which related to apoptosis were observed by the TUNEL analysis. As shown in Figure 5, compared to the positive control treated with *DNase* I (Figure 5A), negative control without rTdT Enzyme incubation (Figure 5B), and the saline groups (Figure 5C), the JEG3 cells in the DGL/CSA-PNPs-treated group showed significantly increased apoptosis (*P* < 0.01, Figure 5D).

### 2.8. DGL/CSA-PNPs Induce the JEG3 Apoptosis through Caspase-3 and P53

The activation of PARP and the release of mitochondrial factor were found both in JEG3 cells in vitro (Figure 6) and in JEG3 tumor *in vivo* (Figure 7). The activated PARP and AIF were clearly observed in the DGL/CSA-PNPs-treated groups compared to the control cells (Figure 6A,B). The expression of PARP and AIF in the JEG3 tumors treated with the DGL/CSA-PNPs as significantly higher than that in the control tumors (Figure 7A,B).

To investigate whether the DGL/CSA-PNPs promote cancer-cell apoptosis via the mitochondria-cytochrome c pathway, including caspase-3, caspase-9, bcl-2 and bax, the protein expression of key regulators was analyzed in JEG3 cells and the tumors treated with DGL/CSA-PNPs. The active caspase-3 and caspase-9 bax were significantly induced in the DGL/CSA-PNPs-treated group compared to the control JEG3 cells or tumors (Figure 6A,B and Figure 7). Meanwhile, the expression of bcl-2 significantly decreased in the DGL/CSA-PNPs-treated group (Figure 6A,B, Figure 7, Appendix A). Additionally, the DGL/CSA-PNPs treatment significantly decreased the expressions of IAPs proteins in the JEG3 cells compared to the control group (Figure 6A,B and Figure 7A,B). Simultaneously, after the treatment with the DGL/CSA-PNPs, the expression level of P53 was significantly up-regulated compared to the untreated cells (Figure 6A,B and Figure 7A,B). However, after treatment the cells with permeable caspase-3 inhibitor Z-D(OMe)E(OMe)-FMK and inhibitor of p53-mediated apoptosis pifithrin-μ, the up-regulation of the expression levels of caspase-3 protein and P53 protein induced by DGL/CSA-PNPs in JEG3 cells were blocked (Figure 6C,D).

## 3. Discussion

Previous research has reported that prodigiosin-producing strains include *Serratia, Streptomyces, Hahella,* and *Vibrio* [27,28]. Strain HDZK-BYSB107 belongs to *Serratia*, which was isolated from Port-Orford-Cedar (POC), *C. lawsoniana*. It is noteworthy that strain HDZK-BYSB107, which we identified as *Serratia marcescens* subsp. *lawsoniana*, is an endophytic bacterium from *C. lawsoniana*. This is the first report of a prodigiosin-producing strain from *C. lawsoniana*. As prodigiosin is a natural product, the study of its biosynthetic pathways could uncover more potent derivatives, leading us closer to the ultimate goal of finding improved chemotherapeutics for the treatment and management of cancer. The prodigiosin and its biosynthesis methods have been developed for several years since it was first identified as a significant anticancer drug with pronounced features. Most of the available anticancer drugs on the market are chemically synthesized and have significant side effects [2], so this made the microbial original prodigiosin a promising alternative. The cytotoxic activity of prodigiosin against a large number of human cancer cells and relatively lower toxicity toward nonmalignant cells [3,5]. To further increase the drug-delivery efficiency in cancer therapy, the efforts to create novel derivatives of prodigiosin and analogs with pharmaceutical importance through chemical synthesis and biosynthesis are ongoing and play a critical role in cancer therapy. In this study, we investigated the anticancer efficiency of this bacterial prodigiosin on JEG3 cells. Our results indicate the potential applications of HDZK-BYSB107 prodigiosin, especially anticancer potency, and suggest that bacterial prodigiosin is an alternative therapeutic drug for tumors.

Choriocarcinoma, a type of gestational trophoblastic neoplasia, can follow any form of pregnancy, including term pregnancy, abortion, and ectopic gestation. Choriocarcinoma is a highly malignant neoplasm resulting from the malignant transformation of proliferating trophoblastic cells [29,30]. We developed a targeted-delivery drug carrier for choriocarcinoma recently, with CSA-binding peptide-coated nanoparticles composed of PLGA, soybean lecithin, and DSPE-PEG-COOH [20]. However, the loading efficiency and encapsulation efficiency are relatively low. Recently, DGL nanoparticles emerged as an effective vehicle and a new kind of synthetic polymers for the delivery of drugs in intravenous administration [31,32,33]. Therefore, in this study, we synthesized DGL formulations of DGL/CSA-PNPs, and they have a significantly higher anticancer activity against JEG3 cells compared to HDZK-BYSB107 prodigiosin alone, *in vitro*, as well as JEG3 tumor model *in vivo*.

Although we are unable to fully understand the mechanisms of the multiple biological activities of prodigiosin, many studies have suggested that prodigiosin can induce apoptosis in many cancer cells [34]. In our study, JEG3 cells and tumors were used to further evaluate the anticancer effect and mechanism of the DGL/CSA-PNPs on JEG3 cells. Four possible mechanisms were suggested for DGL/CSA-PNPs: as DNA cleavage agents, as pH modulators, as cell cycle inhibitors, and as mitogen-activated protein kinase regulators [35]. Studies indicate that apoptosis mainly depends on the apoptotic pathways of mitochondria-cytochrome c; cytochrome c released by mitochondria can form a complex with caspase-9 and Apaf-1 and activate caspase-3 to start a caspase cascade reaction, which ultimately induces apoptosis [36,37]. In the present study, the apoptosis induced by the DGL/CSA-PNPs was confirmed in JEG3 cells or tumors, and the apoptotic response was mainly through the mitochondria-cytochrome c pathway. DGL/CSA-PNPs induced the caspase-9 and caspase-3 activation, and the PARP was subsequently activated. On the other hand, we know that the anti-apoptotic pathway includes bcl-2 and bcl-xl, while others are pro-apoptotic, such as bax, which are closely associated with tumors [38,39]. In this study, we have also confirmed that, for JEG3 cells and tumors during the DGL/CSA-PNPs treatment, bax protein expression increased, while bcl-2/bax decreased. As is well-known, inhibitor of apoptosis family can directly inhibit apoptotic caspases. To date, six members were found in IAPs family of humans, including XIAP, cIAP1, cIAP2, survivin, NIAP, and BRUCE. Research on mechanism of IAPs-induced apoptosis inhibition showed the essential role of IAPs in apoptosis inhibition. During the cell cycle, p53 protein repairs damaged DNA by preventing cells in the G1 phase to transition into the S-phase [40]. Consistent with other reports, our findings also indicated the apoptosis of cancer cells in a dose- and time-independent manner [41]. Compared with the data of *in vivo* antitumor analysis, it can be concluded that the JEG3 cell death induced by the DGL/CSA-PNPs can’t be explained only by caspase-dependent pathways, but that other mechanisms also contribute to the observed JEG3 cell death.

Human-choriocarcinoma-targeted DGL/CSA-PNPs were successfully synthesized and complexed with HDZK-BYSB107-prodigiosin-yielding nanoparticles. The mechanism of DGL/CSA-PNPs mediated JEG3-cells-targeting is affected by plCSA-BP, a placenta CSA-binding peptide (plCSA-BP), which can target most of human cancer cells by binding to plCSA on the cell membrane and was similar to protein VAR2CSA. Analysis of anticancer efficiency *in vivo* showed that the DGL/CSA-PNPs were efficient enough for choriocarcinoma-targeted drug delivery. It should be believed that the DGL/CSA-PNPs have great application prospects as the polymer vector for relatively safe and efficient JEG3-cells-targeted drug delivery. Furthermore, our results suggested that the DGL/CSA-PNPs possess excellent cancer-targeting capacity and superior antitumor efficacy.

## 4. Materials and Methods

### 4.1. Animals

A total of thirty female CD-1 mice and forty female NU/NU nude mice (Balb/c, 8 weeks old) were purchased from the Medical Experimental Animal Center of Guangdong Province, and maintained under cycles of 12 h of light and 12 h of dark, with free access to food and water. Care of laboratory animals and experiments on animals were done in accordance with the protocols approved by the Animal Ethics Committee of the Shenzhen Institutes of Advanced Technology, Chinese Academy of Sciences (SIAT), China (SIAT-IRB-170406-YYS-FXJ-A0350, approval date: April 6, 2017), for Animal Care and Use.

### 4.2. Reagents and Antibodies

Dendrigraft poly-L-lysines (DGL) (containing 123 primary amino groups, generation 3) were purchased from COLCOM (Montpellier Cedex, France). The α-Malemidyl-*N*-hydroxysuccinimidyl polyethylene glycol (NHS-PEG-MAL, MW 3500) was obtained from JenKem Technology Co., Ltd. (Beijing, China). The BODIPY fluorophore (4,4-difluoro-5,7-dimethyl-4-bora-3a,4a-diaza-s-indacene- 3-propionic acid, sulfosuccinimidyl ester, sodium salt) and 4,6-diamidino-2-phenylindole (DAPI) were purchased from Molecular Probes (Eugene, OR, USA). Dulbecco’s Modified Eagle’s Medium (DMEM) medium (Life Technologies, Inc., Eggenstein, Germany), supplemented with 10% bovine serum (FBS) (Gibco, Tulsa, OK, USA), and 0.25% (w/v) trypsin solution were purchased from Gibco (Tulsa, OK, USA). Primary antibodies of rabbit anti-caspase-3 antibody, rabbit-caspase-9 antibody, rabbit anti-p53 antibody, rabbit anti-Bcl-2 antibody, rabbit anti-Bax antibody, rabbit anti-AIF antibody, rabbit anti-cIAP1 and rabbit anti-cIAP2 antibody, rabbit anti-PARP antibody, and rabbit anti-XIAP antibody were obtained from abcam (Abcam, Cambridge, MA, USA). Anti-mouse and anti-rabbit secondary antibody horseradish peroxidase conjugates were obtained from Cell Signaling Technology. The cell permeable caspase-3 inhibitor Z-D(OMe)E(OMe)-FMK and inhibitor of p53-mediated apoptosis pifithrin-μ were purchased from abcam (Abcam, Cambridge, MA, USA). The BCA (bicinchoninic acid) protein assay kit was obtained from Pierce (Rockford, IL, USA). Novex precast Tris-glycine gels were purchased from Bio-Rad Laboratories Inc. (Hercules, CA, USA).

### 4.3. Prodigiosin from Strain HDZK-BYSB107

Strain HDZK-BYSB107 can produce prodigiosin and is stored in the China Center for Type Culture Collection, under patent number CCTCC NO 209163. The prodigiosin used in the study is obtained from the strain [42].

### 4.4. In Vivo Safety of DGL

A total of thirty female CD-1 mice were randomly divided into six groups and treated as follows: Group 1 mice were injected with 0.9% saline, group 2–6 mice (*n* = 5 each group) were injected with 50, 100, 200, 500, and 1000 μg/kg of DGLs, every two days, for 30 days, through tail vein, respectively. Two days after the last injection, the mice were anesthetized, and then the lungs, kidney, and liver were collected from each, for histology examination.

### 4.5. Synthesis of the plCSA-BP Targeted DGL Derivatives Nanoparticles

The synthetic route for targeted prodigiosin encapsulated in plCSA-BP-modified DGL (EDVKDINFDTKEKFLAGCLIVSFHEGKC) nanoparticles (DGL/CSA-PNPs) and the scrambled peptide (SCR, EVDNDKKLGLVFEKDKIFTEFACISHCG) conjugated nanoparticles (DGL/SCR-PNPs) were illustrated in (Scheme 1). Briefly, DGL (containing 123 primary amino groups, generation 3, COLCOM (Montpellier Cedex, France)) reacted with the α-malemidyl-N-hydroxysuccinimidyl polyethylene glycol (NHS-PEG-MAL, MW 3500, JenKem Technology Co., Ltd. (Beijing, China)) at molar ratio 1:5 (mol/mol), in PBS (pH 8.0), for 2 h, at room temperature. The primary amino groups on the surface of DGL specifically reacted with the NHS groups of the bifunctional PEG derivative. The resulting DGL-PEG NPs (DGL-NPs) were purified by ultrafiltration, using a 5 kDa molecular weight cutoff membrane, and the buffer was then changed to PBS (pH 7.0). Then, DGL-NPs reacted with plCSA-BP or SCR that was synthesized from China Peptides Co., Ltd. (Shanghai, China) with a molar ratio of 1:2 (mol/mol, DGL to peptides), in PBS (pH 7.0), for 24 h, at room temperature. The MAL groups of DGL-NPs specifically reacted with the thiol groups of plCSA-BP or SCR peptide, yielding the DGL-PEG-CSA NPs (DGL/CSA-NPs) carrier (Scheme 1A). After purifying by ultrafiltration, using a 5 kDa molecular weight cutoff membrane, the characteristics of DGL/CSA-NPs were analyzed by transmission electron microscopy (Hitachi Ltd., Tokyo, Japan) and Zetasizer Nano 2000 (Malven, England).

To visualize the biding of the nanoparticles, BODIPY was used to label the DGL and synthesize nanoparticles. Briefly, DGL first reacted with BODIPY fluorophore (4, 4-difluoro-5, 7-dimethyl-4-bora-3a, 4a-diaza-s-indacene-3-propionic acid, sulfosuccinimidyl ester, sodium salt, Eugene, OR, USA) in 100 mmol/l NaHCO_3_, for 1 h, at room temperature, and then purified to remove unreacted BODIPY by ultrafiltration through a membrane (cut off×5 kDa). The BODIPY-labeled DGL were used to synthesize different BODIPY-labeled carrier as BODIPY/DGL-PEG-CSA NPs (DGL/CSA-BNPs) and BODIPY/DGL-PEG-SCR NPs (DGL/SCR-BNPs), as described above.

To load the prodigiosin from HDZK-BYSB107 into DGL derivatives, the fresh prodigiosin solution was encapsulated in DGL-NPs, DGL/CSA-NPs, and DGL/SCR-NPs, by an organic/water emulsion-solvent evaporation method (Scheme 1B). First, 4 mg of prodigiosin was dissolved in 1 mL of dimethyl sulfoxide (organic phase), and 20 mg of the DGL-NPs, DGL/CSA-NPs, or DGL/SCR-NPs solution (1 mg/mL in PBS buffer, pH 7.4) was added to the organic phase (primary emulsion). Then, 5 mL of 2% aqueous solution of polyvinyl alcohol (PVA) was added dropwise to the primary emulsion; and the preparation was stirred at 500 RPM/min, for 5 h, at the room temperature. Last, the resulting DGL/CSA-PNPs and DGL/SCR-PNPs were recovered by centrifugation, at 5000 RPM/min, for 10 min, at 4 °C, and then washed three times with sterilized deionized water, centrifuged, and freeze-dried.

### 4.6. Characterization of the DGL/CSA-PNPs and DGL/SCR-PNPs

The ligation efficiency of plCSA-BP on DGL/CSA-PNPs was evaluated by a Pierce BCA Protein Assay Kit (Thermo Inc., Waltham, MA, USA) by following the instructions [43]. The unconjugated plCSA-BP or SCR peptide were removed by ultrafiltration, using a 5 kDa molecular weight cutoff membrane. Thereafter, plCSA-BP or SCR peptide concentration conjugated on DGL/CSA-PNPs and DGL/SCR-PNPs were evaluated by Pierce BCA Protein Assay Kit (Thermo Inc., Waltham, MA, USA). The morphological and surface characteristics of the DGL/CSA-PNPs were examined by transmission electron microscopy (Hitachi Ltd., Tokyo, Japan). Zeta potential and particle size of diameter of the nanoparticles were measured by Zetasizer Nano (Malvern, England) by following the manufacturers’ instructions. Encapsulation efficiency (EE) and loading capacity (LC) were calculated with the following equations:EE= (W_0_ − W_1_)/W _0_ × 100%(1)
LC= (W_0_ − W_1_)/W_N_ × 100%(2)
where W_0_ is the total amount of prodigiosin, W_1_ represents the weight of free prodigiosin, and W_N_ indicates the weight of the nanoparticles. The amount of prodigiosin in the nanoparticles was measured by a Beckman Coulter (Indianapolis, IN, USA) DU 800 UV-Vis spectrophotometer, based on their absorbance at 533 nm.

### 4.7. In Vitro Release Assay

In vitro release assay was carried out to determine the release profiles of prodigiosin from the different nanoparticles in PBS buffer (pH 7.4) and in acetic buffer (pH 5.3), at 37 °C, separately. Briefly, the prodigiosin-loaded nanoparticles (0.2 mg) were put into a dialysis bag and dispersed in 1 mL of PBS buffer or acetic buffer. The dialysis bags were then put into the beaker with 100 mL PBS buffer (pH 7.4) or acetic buffer (pH 5.3) and then shaken at 500 RPM/min, at 37 °C. Then, 1 mL of samples were periodically collected at different time points (0, 2, 4, 6, 8, 10, 12, 24, 36, 48, and 72 h) from the beaker and analyzed by a UV spectrophotometer, at 533 nm. The sample volumes removed were replaced with an equal volume of fresh PBS (pH 7.4). All the measurements for each sample were performed in triplicate, and the mean values were calculated. The release curve of prodigiosin from DGL/CSA-PNPs and DGL/SCR-PNPs were plotted, using release time as the *x* axis and cumulative release amount as the *y* axis.

### 4.8. Cytotoxicity Analysis

The cytotoxicity of DGL was measured by human embryonic kidney cell lines (HEK293) that were purchased from the American Type Culture Collection (Manassas, VA, USA), using the Cell Counting Kit-8 reagent (CCK-8, Dojindo Molecular Technologies, Inc., Kumamoto, Japan). HEK293 cells were seeded in a 96-well plate, at 5 × 10^3^ cells per well, in 100 μL of DMEM medium containing 10% FBS and 1% penicillin-streptomycin. Cells achieving 70%–80% confluence were exposed to 100 μL of different concentrations (5, 10, 20, 50, and 100 μg/mL) of nanoparticles (DGL, DGL/CSA-NPs, and DGL/CSA-PNPs) for 3 h, at 37 °C, in an incubator under a humidified atmosphere containing 5% CO_2_. The wells which contained DMEM alone or CCK-8 reagent and DMEM medium served as controls. Then, 10 μL of CCK-8 solution was added into each well, then incubated for 3 h, at 37 °C, in a humidified 5% CO_2_ atmosphere. OD_450_ was measured by a microplate reader (Bio-TEK, MQX200R Power Wave^TM^ XS, Mannedorf, Switzerland), and the survival rate of cells was calculated as follows:Survival rate (%) = [(As − Ab)]/(Ac − Ab)] ×100%(3)
where As represents the OD_450_ of test wells (containing cell medium, CCK-8 and DGL or DGL/CSA-NPs or DGL/CSA-PNPs), Ab represents the OD_450_ of blank wells (containing cell medium alone), and Ac represents the OD_450_ of control wells (containing cell medium and CCK-8). Cell spreading was estimated according to the round and non-round shape ratio, by observation under a phase-contrast microscope (Olympus CKX 41; Olympus, Hachioji, Japan).

### 4.9. In Vitro Specific Binding Assay

The human choriocarcinoma cell line (JEG3) was used to evaluate the specific binding ability of the plCSA-BP-coated nanoparticles and exclude the endocytosis effect of the cells. JEG3 cells (obtained from the Cell Resource Center of Shanghai College of Health Sciences, Chinese Academy of Sciences shanghai, China) were cultured with DMEM/F-12 containing 10% fetal calf serum (FBS) and 1% penicillin/streptomycin, on glass coverslips. At 60% confluency, the culture medium was discarded, and the JEG3 cells were incubated with free BODIPY, DGL/CSA-BNPs, and DGL/SCR-BNPs, at a concentration of 5 μg/mL of BODIPY or equivalent in 500 μL serum-free DMEM/F-12 medium, for 1 h, at 4 °C, to exclude autonomous endocytosis of these nanoparticles. After being washed with PBS and replaced with fresh media without BODIPY or nanoparticles, these cells were then transferred to a humidified 5% CO_2_, at 37 °C, for another 30 min, to evaluate the nanoparticles uptake efficiency between DGL/CSA-BNPs and DGL/SCR-BNPs. Then, the cells were directly fixed with 4% paraformaldehyde, for 30 min, at room temperature. Coverslips were mounted with mounting media containing DAPI (Vector Laboratories, CA, USA) and analyzed by laser-scanning confocal microscopy (Leica TCS SP5, Braunschweig, Lower Saxony, Germany).

### 4.10. In Vitro Uptake Efficiency Assay

Then, 70%–80% confluency JEG3 cells in 6-well plates were incubated with free BODIPY, DGL/CSA-BNPs and DGL/SCR-BNPs, in serum-free DMEM/F-12 medium, for 1 h, at 4 °C, and subsequently transferred to a humidified 5% CO_2_, at 37 °C, for another 30 min, after being washed with PBS and replaced with fresh media without BODIPY or nanoparticles. The cells were then dislocated and used to quantify cellular uptake of different drug formulations and plCSA or SCR peptide binding by a FACS Calibur (BD Biosciences). Meanwhile, JEG3 cells incubated with DMEM/F12 medium served as the control.

### 4.11. IVIS In Vivo Imaging

The anticancer effect of the free prodigiosin and DGL nanoparticle formulations were evaluated in JEG3 cells bearing nude mice. To achieve this goal, the JEG3 cell lines were transduced by lentivirus-expressing firefly luciferase and tomato produce, the firefly luciferase and the tomato positive cell line of JEG3-Fluc-Tomato, for easier daily monitoring of tumor growth in live animal models [44]. Briefly, 1 × 10^6^ JEG3 cells in 100 μL of PBS were subcutaneously injected to the right upper flanks of a female nude mouse to establish the subcutaneous tumor model. Simultaneously, sixty female mice (BALB/c, 8 weeks old) were randomly divided into five groups (*n* = 4): physiological saline, free prodigiosin, DGL-PNPs, DGL/SCR-PNPs, and DGL/CSA-PNPs group, respectively. Each mouse received 1.5 mg/kg prodigiosin or equivalent every two days, for 18 d, via tail-vein injection, starting from 2 days post the JEG3-cells injection.

The animals were imaged on alternate days, from 1 d to 18 d after treatments injection. Briefly, each nude mouse was intraperitoneally administered with D-luciferin (150 mg/kg body weight, Gold Biotechnology, Inc., Saint Louis, MO, USA) in 100 μL of PBS by intraperitoneal injection, 10 min prior to imaging. Then, the bioluminescence intensity in the mice was scanned using a Xenogen IVIS spectrum (Caliper, Newton, MA, USA). *In vivo* tumor signal quantification is presented as an absolute signal in reference to the signal of the right upper flanks of the control mouse. Data analysis was performed using the Living Image Software (Caliper, Newton, MA). The mice were anesthetized if the tumor volume grew up to 1000 mm^3^ or by the end of the treatments. The body weight of mice was measured every three days, and the size of the tumor was measured and calculated by the following equation: V tumor = LWH/2 (L: tumor length, W: tumor width, H: height) every day, as described before [45]. The survival rate was subsequently evaluated, respectively.

### 4.12. Analysis of Apoptosis Induced by the Prodigiosin and DGL/CSA-PNPs

To test the apoptotic effect induced by prodigiosin from strain HDZK-BYSB107 with the different treatments *in vitro*, JEG3 cells in 6-well plate were treated with the prodigiosin from strain HDZK-BYSB107 or prodigiosin standard (Sigma-Aldrich) in different concentrations (0, 5, 10, and 50 μg/mL), for 24 h, at 37 °C. Then, the cells were incubated for another 24 h after treatment, trypsinized and stained, according to the Annexin V/FITC/PI apoptosis assay kit (Solarbio, China), and analyzed by FACS flow Cytometer (BD Biosciences, San Diego, CA, USA) and Diva software. In another group, cells were treated with the cell permeable caspase-3 inhibitor Z-D (OMe) E (OMe)-FMK (Abcam Inc., Cambridge, MA, USA) and inhibitor of p53-mediated apoptosis pifithrin-μ (Abcam Inc, Cambridge, MA, USA), in order to verify the caspase-3 and P53 protein expression alteration in DGL/CSA-PNPs-treated JEG3 cells. Cells were collected for protein extraction after the treatment.

In the meantime, the JEG3 tumors that were collected from the DGL/CSA-PNPs-treated nude-mice tumor model were embedded in optimal cutting temperature compound (TISSUE-TEKs 4583, Sakura Finetek Inc., Torrens, CA, USA) and 10 μm thick sections were cut and fixed with 4% paraformaldehyde (PFA), for 30 min, at room temperature, before staining with DeadEnd^TM^ Colorimetric TUNEL System for apoptosis analysis (Promega Inc., Madison, WI, USA), following the manufacturer’s protocol. Images were acquired by a light microscope (Olympus BX53, Japan).

### 4.13. Real-Time PCR

Total RNA of the JEG3 cells and JEG3 tumors were extracted using TRIzol^®^ reagent (Takara, Dalian, China). The cDNAs were reverse-transcribed from total RNAs, by the using GoScript™ Reverse Transcription System (Promega, Madison, WI, USA). Real-time PCR reactions were carried out using a SYBR^®^ Premix Ex Taq Kit (DNing, China). To normalize target gene expression, β-actin was used as an internal control. The cycle threshold data were obtained, and the relative expression levels were analyzed using the 2^−ΔΔCt^ method [46].

### 4.14. Western Blot Assay

Total protein from JEG3 and JEG3 tumors from different treatment groups was run in PAGE (sodium dodecyl sulfate polyacrylamide gel electrophoresis) gel and transferred to a PVDF (polyvinylidene fluoride) membrane and probed with the following primary antibodies, for 12 h, at 4 °C: caspase-3, caspase-9, p53, bax, bcl-2, AIF, cIAP 1 and cIAP 2 (Abcam Inc., Cambridge, MA, USA, 1:1000), X-linked inhibitor of apoptosis protein (XIAP) (Santa Cruz Biotechnology Inc., USA, 1:1000), poly (ADP-ribose)-polymerases (PARPs), and β-actin (Abcam Inc., Cambridge, MA, USA, 1:1000). Then, the blots were incubated with HRP (horseradish peroxidase)-conjugated goat anti-rabbit secondary antibody for 1 h and washed. Following detection with FluorChem E Chemiluminescent Western Blot Imaging System (Cell Biosciences, Santa Clara, CA, USA), the blots were quantified by densitometry analysis, using Image J software (National Institutes of Health, Bethesda, MD, USA). All Western blot data shown are representative of at least three independent experiments.

### 4.15. Immunofluorescence and Immunocytochemistry Assay

JEG3 cells on glass cover slides and 10 μm of JEG3-tumor frozen sections were fixed in 4% PFA, blocked in 1% BSA (Bovine Serum Albumin) for 30 min, and then incubated with caspase-3, bcl-2 and p53 primary antibodies for 12 h, at 4 °C. Cells were subsequently incubated with Alexa488 fluorophore conjugated secondary antibodies (Molecular Probes, Eugene, OR, USA) for 30 min. Coverslips were mounted in mounting media containing DAPI (4′,6-diamidino-2-phenylindole) and analyzed by laser-scanning confocal microscopy (Leica TCS SP5, Braunschweig Wetzlar and Mannheim, Lower Saxony, Germany). The following immunocytochemistry assay was performed according to previous report [42].

### 4.16. Statistical Analysis

All experimental results were expressed as mean values ± standard deviation (SD). All experiments were repeated at least three times, with triplicated samples in each experiment. To estimate the statistical significance of differences between two treatment groups, we used unpaired Student’s t-tests to calculate two-tailed *P*-values. One-factor analysis of variance (ANOVA) was employed to evaluate the statistical differences among more than two groups, with SPSS 19.0 software. The difference between groups with *P* < 0.05 and *P* < 0.01 was considered to be statistically significant.

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
