# Peer review of "Targeted Delivery Prodigiosin to Choriocarcinoma by Peptide-Guided Dendrigraft Poly-l-lysines Nanoparticles"

_ijms, 2019, doi:10.3390/ijms20215458_

Round 1

Reviewer 1 Report

In this study, the author newly constructed dendrigraft poly-L-lysines (DGL) modified with placental chondroitin sulfate (CSA)-binding peptide. Prodigiosin was encapsulated in the DGL modified by the plCSA-BP nanoparticles (DGL/CSA-NPs). In in vitro study, the DGL/CSA-NPs showed significantly higher cellular uptakes than DGL/SCR-NPs which was constructed with scrambled peptide. After intravenous administration into JEG3 cells bearing nude mice, DGL/CSA-NPs containing prodigiosin induced apoptosis in the tumor and showed significantly higher anticancer effect than free prodigiosin and DGL/SCR-NPs containing prodigiosin. There are few reports about nanoparticles containing prodigiosin. Therefore, this study would be available for the readers of the International Journal of Molecular Sciences. However, several points should be addressed and revised before publication. This manuscript will be accepted after major revisions.

Major

Comment 1:

In vivo distribution of free prodigiosin and DGL/CSA-PNPs should be determined.

Comment 2

In the confocal microscopy image (Fig. 3A), DGL/CSA-BNPs seem to be taken into the cell regardless of 4 ℃ incubation. If the endocytosis was inhibited by 4 ℃ incubation, the author should mention the uptake mechanism of DGL/CSA-BNPs.

Comment 3:

In the Fig3C, 4B-D, 6D, student’s t-test is not suitable for statistical analysis. The author must analyze those data using ANOVA with post-hoc test such as Bonferroni test, Fisher's LSD test, Dunnet test, and so on.

Minor

Comment 1:

In the line 194 to 196, “The active caspase-9 and caspase-9, bax were significantly・・・” might be “The active caspase-3 and caspase-9, bax were significantly・・・”.

Comment 2:

In the line 204, “(Figure 7C-D)” might be “(Figure 6C-D)”

Comment 3:

In the line 290, what is Malemidyl--N-hydroxysuccinimidyl polyethyleneglycol (NHS-PEG-MAL) meaning.

Comment 4

In the line 462-463, “cDNAs were reverse transcribed from total RNAs according to the manufacturer’s instruction”. There is no information about a reagent using for RT reaction.

Author Response

Reviewer #1

General comments:

In this study, the author newly constructed dendrigraft poly-L-lysines (DGL) modified with placental chondroitin sulfate (CSA)-binding peptide. Prodigiosin was encapsulated in the DGL modified by the plCSA-BP nanoparticles (DGL/CSA-NPs). In vitro study, the DGL/CSA-NPs showed significantly higher cellular uptakes than DGL/SCR-NPs which was constructed with scrambled peptide. After intravenous administration into JEG3 cells bearing nude mice, DGL/CSA-NPs containing prodigiosin induced apoptosis in the tumor and showed significantly higher anticancer effect than free prodigiosin and DGL/SCR-NPs containing prodigiosin. There are few reports about nanoparticles containing prodigiosin. Therefore, this study would be available for the readers of the International Journal of Molecular Sciences. However, several points should be addressed and revised before publication. This manuscript will be accepted after major revisions.

RESPONSE: Special thanks for your good comments and the valuable suggestions for improving our work. We have tried our best to address these comments and suggestions and carefully revised the manuscript as you have instructed. We believe that the revised version would meet the high quality requirements of the journal, and we would also be glad to further revise this manuscript if the need arises.

Major comments: 

1. In vivodistribution of free prodigiosin and DGL/CSA-PNPs should be determined.

RESPONSE: Special thanks for pointing this out. Actually, we mainly focus our research on the targeting therapy of choriocarcinoma by placental chondroitin sulfate (CSA)-binding peptide (plCSA-BP), and the antitumor mechanism of this kinds of DGL formulations. And the targeted delivery efficiency of the plCSA-BP guided nanoparticles has been validated with other formulations (Refer Baozhen Drug delivery). That’s why we have not done the in vivo distribution investigation. Thank you very much!

2. In the confocal microscopy image (Fig. 3A), DGL/CSA-BNPs seem to be taken into the cell regardless of 4℃ incubation. If the endocytosis was inhibited by 4℃ incubation, the author should mention the uptake mechanism of DGL/CSA-BNPs.

RESPONSE: Thanks for the valuable comments. We are very sorry for our incorrect expression. As shown on Line 134 on Page 7. The cells were incubated with DGL/SCR-BNPs in serum-free DMEM/F-12 medium for 1 h at 4o C to exclude autonomous endocytosis of these nanoparticles. These cells were then transferred to a humidified 5% CO2 atmosphere at 37o C for another 30 min to evaluate the nanoparticles uptake efficiency between DGL/CSA-BNPs and DGL/SCR-BNPs, however, their major uptake mechanism was phagocytosis.

3. In the Fig3C, 4B-D, 6D, student’s t-test is not suitable for statistical analysis. The author must analyze those data using ANOVA with post-hoc test such as Bonferroni test, Fisher's LSD test, Dunnet test, and so on.

RESPONSE: Thanks for your good comments. As the reviewer suggested, those data of Fig. 3C, Fig. 4B-D, Fig. 6D in the revised manuscript were analyzed by using One-way ANOVA on Line 148 on Page 8, Line 168 on Page 9, Line 217 on Page 12 and Lines 494-496 on Page 26 in the revised manuscript. Thank you very much!

Minor comments: 

1. In the line 194 to 196, “The active caspase-9 and caspase-9, bax were significantly・・・”might be “The active caspase-3 and caspase-9, bax were significantly・・・”.

RESPONSE: We are sorry for the typo. As the reviewer suggested that we have changed the "caspase-9" to "caspase-3" on Line 200 on Page 11 in the revised manuscript. Thank you very much!

2. In the line 204, “(Figure 7C-D)” might be “(Figure 6C-D)”.

RESPONSE: Sorry for the type, it has been changed as the reviewer suggested on Line 209 on Page 11 in the revised manuscript. Thank you very much!

3. In the line 290, what is Malemidyl--N-hydroxysuccinimidyl polyethyleneglycol (NHS-PEG-MAL) meaning.

RESPONSE: Special thanks for pointing this out. The sentence of “Malemidyl--N-hydroxysuccinimidyl polyethyleneglycol (NHS-PEG-MAL)” was deleted in the revised manuscript.

4. In the line 462-463, “cDNAs were reverse transcribed from total RNAs according to the manufacturer’s instruction”. There is no information about a reagent using for RT reaction.

RESPONSE: Thanks for pointing out the valuable comments. As the reviewer suggested that we have elaborated the details of the reagent using for RT reaction on Lines 465-466 on Page 25 in the revised manuscript. Thank you very much!

Reviewer 2 Report

This work reports a new approach based on dendrigraft poly-L-lysines (DGL) modified with plCSA-BP (a 28 amino acids placental CSA-binding peptide from VAR2CSA) for site specific targeting of choriocarcinoma. The work is innovative and of high scientific and technical quality. Prodigiosin has been selected as chemotherapeutic compound to be loaded into plCSA-BP nanoparticles.  Minor suggestions are given for improvement of the work. Please clarify the statement: “The prodigiosin and its biosynthesis [has] been developing for several years” and edit it for better use of English. Scheme 1 should be provided with better quality and sequentially provided as figure. Dimethyl sulfoxide is an organic solvent, not an oil; for accuracy the terminology should be revised in the entire manuscript and the phase called organic (not oil phase). Section 4.6. states that the binding efficiency will follow the instructions however not mentioning from whom (e.g. supplier?, reference?). The size of nanoparticles that has been recorded is the mean particle size of the hydrodynamic diameter? If so, this should be clearly stated. English revision is strongly recommended (e.g. “Studies [indicates]”; “an [alternate] tumor”.

Author Response

Reviewer #2

General comments:

This work reports a new approach based on dendrigraft poly-L-lysines (DGL) modified with plCSA-BP (a 28 amino acids placental CSA-binding peptide from VAR2CSA) for site specific targeting of choriocarcinoma. The work is innovative and of high scientific and technical quality. Prodigiosin has been selected as chemotherapeutic compound to be loaded into plCSA-BP nanoparticles. Minor suggestions are given for improvement of the work.

RESPONSE: Special thanks for your good comments and valuable suggestions for improving our work. We have tried our best to address these comments and suggestions and carefully revised the manuscript as you have instructed. We believe that the revised version would meet the high quality requirements of the journal. Thanks again!

 Specific comments:

1. Please clarify the statement: “The prodigiosin and its biosynthesis [has] been developing for several years” and edit it for better use of English.

RESPONSE: Thanks for pointing this out. As the reviewer suggested that we have edited the sentence of on Line 232 on Page 13 in the revised manuscript.

2. Scheme 1 should be provided with better quality and sequentially provided as figure. Dimethyl sulfoxide is an organic solvent, not an oil; for accuracy the terminology should be revised in the entire manuscript and the phase called organic (not oil phase).

RESPONSE: As the reviewer suggested, the quality of Scheme 1 was improved as figure. Furthermore, as the reviewers suggested that we have changed the “oil” to “organic” on Line 351, 353 on Page 19 and Line 354 on page 20 in the revised manuscript. Thank you very much!

. Section 4.6. states that the binding efficiency will follow the instructions however not mentioning from whom (e.g. supplier?, reference?).

RESPONSE: We have cited the reference on Line 361 on Page 20 in the revised manuscript. Thank you very much!

Reference:

Walker, J.M. The bicinchoninic acid (BCA) assay for protein quantitation. Methods Mol Biol. 1994,32, 5-8.

4. The size of nanoparticles that has been recorded is the mean particle size of the hydrodynamic diameter? If so, this should be clearly stated. English revision is strongly recommended (e.g. “Studies [indicates]”; “an [alternate] tumor”.

RESPONSE: Thanks for the valuable comments. The particle size refers to diameter. They were clearly stated on Line 100 Page 100, Lines 113-115 on Page 6 and Line 366 on page 20 in the revised manuscript. As you suggested, the English was improved in the revised manuscript.

Round 2

Reviewer 1 Report

Minor points have been improved, however, several responses about major comments seem to be not enough.

1. In vivo distribution of free prodigiosin and DGL/CSA-PNPs should be determined.

RESPONSE: Special thanks for pointing this out. Actually, we mainly focus our research on the targeting therapy of choriocarcinoma by placental chondroitin sulfate (CSA)-binding peptide (plCSA-BP), and the antitumor mechanism of this kinds of DGL formulations. And the targeted delivery efficiency of the plCSA-BP guided nanoparticles has been validated with other formulations (Refer Baozhen Drug delivery). That’s why we have not done the in vivo distribution investigation. Thank you very much!

Authors should mention about in vivo distribution of the plCSA-BP guided nanoparticle in tumor bearing mice and cite the previous report in the manuscript.

2. In the confocal microscopy image (Fig. 3A), DGL/CSA-BNPs seem to be taken into the cell regardless of 4℃ incubation. If the endocytosis was inhibited by 4℃ incubation, the author should mention the uptake mechanism of DGL/CSA-BNPs.

RESPONSE: Thanks for the valuable comments. We are very sorry for our incorrect expression. As shown on Line 134 on Page 7. The cells were incubated with DGL/SCR-BNPs in serum-free DMEM/F-12 medium for 1 h at 4o C to exclude autonomous endocytosis of these nanoparticles. These cells were then transferred to a humidified 5% CO2 atmosphere at 37o C for another 30 min to evaluate the nanoparticles uptake efficiency between DGL/CSA-BNPs and DGL/SCR-BNPs, however, their major uptake mechanism was phagocytosis.

The comment is in apparent conflict with the manuscript (line 405-409), therefore, authors must revise those sentences.

Author Response

Reviewers' comments:

Reviewer #1

General comments:

Minor points have been improved, however, several responses about major comments seem to be not enough.

RESPONSE: Special thanks for your good comments and the valuable suggestions for improving our work. We have tried our best to address these major comments and carefully revised the manuscript as you have instructed. We believe that the revised version would meet the high quality requirements of the journal, and we would also be glad to further revise this manuscript if the need arises.

Special comments:

Authors should mention aboutin vivodistribution of the plCSA-BP guided nanoparticle in tumor bearing mice and cite the previous report in the manuscript.

RESPONSE: Thanks for your good comments. The details about in vivo distribution of the plCSA-BP guided nanoparticles in tumor bearing mice have been reported in our previous work. This can be seen on Lines 158-160 on Page 5 in the revised manuscript. Thank you very much!

Reference:

Zhang, B.Z.; Cheng, G.G.; Zheng, M.B.; Han, J.Y.; Wang, B.B.; Li, M.X.; Chen, J.; Xiao, T.X.; Zhang, J.; Cai, L.T.; et al. Targeted delivery of doxorubicin by CSA-binding nanoparticles for choriocarcinoma treatment. Drug Deliv. 2018, 25, 461-471.

The comment is in apparent conflict with the manuscript (line 405-409), therefore, authors must revise those sentences.

RESPONSE: Thanks for the valuable comments. As you suggested, the sentences on Lines 422-425 on Pages 12-13 were carefully revised in the revised manuscript. Thank you very much!

Round 3

Reviewer 1 Report

The manuscript has been revised well.